# PLAYING FOR YOU: TEXT PROMPT-GUIDED JOINT AUDIO-VISUAL GENERATION FOR NARRATING FACES USING MULTI-ENTANGLED LATENT SPACE

## ABSTRACT

We present a novel approach for generating realistic speaking and talking faces by synthesizing a person's voice and facial movements from a static image, a voice profile, and a target text. The model encodes the prompt/driving text, the driving image, and the voice profile of an individual and then combines them to pass them to the multi-entangled latent space to foster key-value pairs and queries for the audio and video modality generation pipeline. The multi-entangled latent space is responsible for establishing the spatiotemporal person-specific features between the modalities. Further, entangled features are passed to the respective decoder of each modality for output audio and video generation. Our experiments and analysis through standard metrics demonstrate the effectiveness of our model. All model checkpoints, code, and the proposed dataset can be found at: https://github.com/Playing-for-you.

## 1 INTRODUCTION

AI-generated real-time audio-video multimedia communication by rendering realistic human talking faces has recently drawn massive attention[1,2]. Such technology is promising in various applications such as digital communication, aiding communication with individuals with impairments, designing artificial instructors, and developing interactive healthcare (Xu et al., 2024b; Gan et al., 2023). In such applications, generating realistic and real-time speech and visual content simultaneously is a key requirement. Therefore, in an ideal scenario, given a prompt text along with a face image and the audio profile of an individual, a talking human face would be rendered as output with audio (generated speech) and visual narration according to the prompt text.

Generative AI has emerged as a key area of interest in the computer vision and learning representation community. Although existing approaches have made significant strides, they are constrained by their reliance on generating a single modality (Egger et al., 2020; Kim et al., 2021). For example, current text-to-speech models (TTSM) (Ao et al., 2022; Betker, 2022; Casanova et al., 2024) focus primarily on voice synthesis. Similarly, visual generation techniques i.e. talking face models (TFM) (Ren et al., 2021; Rombach et al., 2022; Siarohin et al., 2020; Zhang et al., 2023a; Xu et al., 2024b;c; Zhang et al., 2023b) aim at face video generation given a text or/and audio or/and image as a prompt. Hence both TTSM and TFM techniques are unsuitable for real-life audio-video multimedia communication scenarios such as audio-visual chatbots, as in such situations both realistic video and speech must be generated synchronously and simultaneously. Few efforts have been made in the literature to merge TTSM and TFM by cascading the pipeline (Wang et al., 2023; Zhang et al., 2022). Additionally, (Jang et al., 2024) made an effort to generate talking face and speaking audio jointly for a specific individual from a prompt text.

Further, these TFM (Chen et al., 2024; Zhang et al., 2019) depend on guidance from defined facial properties from the weakly supervised latent information from the reference modality. As a result,

---

[1] https://www.business-standard.com/technology/tech-news/odisha-television-introduces-lisa-india-s-first-ai-news-presenter-123071000767_1.html

[2] https://www.indiatoday.in/india/story/india-today-groups-ai-anchor-sana-wins-global-media-award-for-ai-led-newsroom-transformation-2532514-2024-04-27

poor lip-synchronization and limited ability to tune an existing audio profile for personalizing the video content lead to generation that is far from being realistic. Moreover, expressiveness in facial dynamics along with subtle nuances for realistic facial behavior needs to simultaneously match with audio content temporally to produce realistic talking faces. Further, such synchronization also depends on individual traits, such as their speech intonation and other covariates. Although they are supposed to be important considerations for realistic speaking and taking faces models (STFM), However, this was not in the scope of existing work on STFM (Jang et al., 2024). Therefore, this gap in the literature motivates us to design a prompt text-guided audio-visual multimodal generative STFM that can jointly generate audio and video, given a reference image and reference audio along with the prompt text as input.

Consequently, in contrast to existing literature (See Figure 1), in this work we introduce a novel multi-modal framework designed to address these limitations by generating highly realistic speech and animations from a combination of prompt text, a driving image, and an audio profile as inputs. Specifically, our framework aims to synthesize videos of a talking human face where the person in the image appears to speak along with the generated voice from the provided text for the given identity. Our method enhances the capabilities of existing pretrained models (Xu et al., 2024b) with an advanced parallel mechanism that leverages both visual and auditory data streams. This parallelism ensures that the synthesized videos not only align the subject's facial movements with the spoken text but also synchronize with the generated personalized voice outputs that correspond to the subject's appearance.

A person-agnostic generalized STFM model must encompass a large appearance and acoustic features variation. Furthermore, extracting such structure infor-

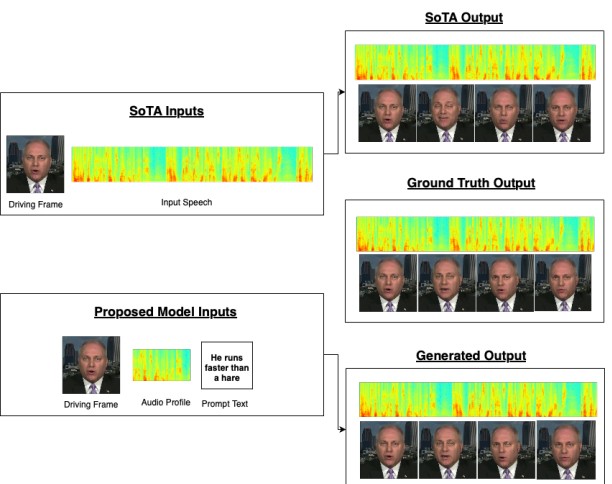

Figure 1: SOTA approaches of talking face generation use a face image as driving frame, with an audio prompt passed as input to the existing model such as Hallo (Xu et al., 2024b), VASA (Xu et al., 2024c) and the proposed model which generates a realistic audio-video synchronous multimodal talking face with face image and audio profile of an individual along with the prompt text.

mation along with the temporal synergy between the audio and video while preserving individual variance requires additional modules to model these complexities. Therefore, we introduce a parallel multiple entanglement in the latent space between the encoding and decoding of different modalities.

Our proposed architecture for STFM contains three main phases (See Figure 2). *Modality encoding phase*, at this stage a heterogeneous personal signature of the audio and video modality, and the driving feature from the text are extracted. The second stage is the *multi-entangled latent space* which glens the spatiotemporal relation and synchronization in the embeddings of the modalities, which further acts as the input to the *decoders phase* i.e the third stage of the proposed architecture. In the second stage, the exchange of information between the key and values (identity information from audio and video extracted from the individual encoders) and queries (driving features from encoded prompt text) are streamlined. To instrument this, an entanglement of the audio and text latent is performed which further entangles with video latent in transformers block and then to a diffusion block. The output of the diffusion block is passed to the video decoder. Similarly, an entanglement of the video and text latent is performed which further entangles with audio latent in a transformer space and passes to a text decoder block and then to the audio decoder. Such entanglements ensure to streamlining of the audio profile and the driving image by linear navigation in the latent space along with the encoded feature from the prompt text. Specifically, the temporal information for both the audio and video generation is constructed by linear displacement of codes in the latent space as per the encoded text prompt. In turn, the model also learns a set of orthogonal motion directions to simultaneously learn the audio and video temporal synergy, by exchanging their linear combination

to represent any displacement in the latent space. To summarize, our key contributions are as follows:

- To the best of our knowledge, the proposed architecture is the first person-agnostic STFM which fosters a text-driven multimodal realistic audio-video synthesis that can be generalized to any identity.

- We design a three-phase architecture which consists of the encoder, multi-entangled latent and decoder phase for audio and video pipeline. The muti-entangled latent space glens the spatiotemporal and synchronisation in the encoder embedding to exchange information between the modality and guided text and help to generate crucial visual and acoustic characteristics based on input profiles.

- With the comprehensive experiments, we demonstrate that the proposed method surpasses the state-of-the-art techniques available for STFM.

## 2 RELATED WORK

Text-to-speech (TTS) technology has seen remarkable progress in recent years, with the development of models that generate highly natural and expressive speech. Modern Text-To-Speech approaches(Casanova et al., 2024; Betker, 2022) leverage sequence-to-sequence architectures to map text directly to speech. Notable models among these are the Tacitron(Wang et al., 2017) and the newer Tacitron2(Shen et al., 2018). These models employ attention mechanisms to convert text sequences into mel-spectrograms. These spectrograms are then passed through neural vocoders like WaveNet(van den Oord et al., 2016) or HiFi-GAN(Kong et al., 2020) to generate high-quality audio waveforms. Other models, such as FastSpeech(Ren et al., 2019) and VITS(Kim et al., 2021), introduce optimizations to improve the speed of speech generation while maintaining or enhancing the naturalness and clarity of the output. Although models have advanced into more complex architectures, the underlying idea behind speech generation remains the same. TortoiseTTS(Betker, 2022) is a modern, expressive TTS system with impressive voice cloning capabilities. This model incorporates a combination of the Auto-Regressive Model, followed by a Diffusion Model(Ho et al., 2020), to convert the input text into mel-spectrogram frames, via discrete acoustic tokens. This model also follows the standard of a vocoder(Univnet)(Jang et al., 2021) for generating the audio from the spectrogram frames. Only a few works have been made in the literature to attend STFM by cascading the pipeline (Wang et al., 2023; Zhang et al., 2022). In (Jang et al., 2024) advancements are made by generating a talking face and speaking audio jointly for a specific individual from a prompt text.

### 2.1 FACE REENACTMENT AND LIP-SYNC MODELS

Recent advancements in face reenactment have enabled realistic video generation by synthesizing facial movements driven by audio inputs. Early models, such as SyncNet(Raina & Arora, 2022), focused on lip synchronization through facial key points and phoneme mapping but struggled with capturing detailed expressions and diverse facial structures. More recent models, such as LipGAN(K R et al., 2019) and Wav2Lip(Prajwal et al., 2020a), leverage GANs to improve lip-sync accuracy and generate more natural facial animations.

The multimodal synthesis of human videos, combining text, audio, and visual inputs, has advanced considerably in recent years. Early approaches focused on audio-driven models that primarily addressed lip-syncing, mapping speech inputs to corresponding facial movements. Models like SyncNet(Raina & Arora, 2022) played a crucial role in establishing baseline synchronization between audio and lip movements. However, these models often lacked expressive, natural face dynamics.

### 2.2 DIFFUSION-BASED LIP-SYNC MODELS

Recent models have extended beyond simple lip-syncing to incorporate emotional expression and natural head motion. Audio2Head(Wang et al., 2021), for example, shifts from keypoint-based methods to a dense mapping of audio features onto facial expressions and head motion, resulting in a more fluid and expressive representation of speech-driven animations. Expressive Audio-driven Talking-heads (EAT)(Gan et al., 2023) enhances this by integrating text and audio as inputs, introducing more dynamic and natural facial expressions synchronized with speech.

The Hallo(Xu et al., 2024b) model builds on these advancements by using attention mechanisms to improve facial reenactment, ensuring smoother transitions and better coherence across diverse speakers. Furthermore, SadTalker(Zhang et al., 2023b) incorporates 3D facial representations, combining both speech and facial dynamics for more realistic head motions and expressive gestures.

FaceChain-ImagineID(Xu et al., 2024a) uses latent diffusion to generate talking faces directly from the only audio input, generating synthetic faces after disentangling the audio to extract aspects like expression, identity and emotion. Other notable works, such as Diffused Heads(Stypułkowski et al., 2023) and DreamTalk(Zhang et al., 2023a), have explored diffusion-based models for video generation, leveraging the success of image-to-video transformations in generating high-quality talking-head videos. These models focus on temporally consistent video generation, addressing fidelity and synchronization across frames.

## 3 METHODOLOGY

We propose a joint learning methodology for the audio, video, and natural language-based text prompts consisting of three main components – namely, **(1)** Encoding phase, **(2)** Entanglement of combined latent space, and **(3)** Decoding phase *i.e.,* Latent conditional generation of synthesized audio-video. Figure 2 illustrates detailed network architecture and roles of different model components to learn and dynamically synthesize audio video on a given source image.

### 3.1 MULTI-MODAL ENCODING PHASE.

We use HiFi-GAN (Kong et al., 2020) and Wav2Vec Encoder (Baevski et al., 2020) to extract high-dimensional embedding vectors from the reference audio. The HiFi-GAN generates a feature embedding $\mathbf{f}_a$ that represents the audio waveform. At the same time, the Wav2Vec encoder produces a secondary set of embedding $\mathbf{f}_s$ capturing semantic audio information. We treat the semantic audio embedding as a direct mapping of the speaker's voice profile. Consequently, the combined features $\mathbf{f}_a \oplus \mathbf{f}_s$ provide a detailed audio profile necessary for driving the lip-sync and facial animations in the synthesized video. The input reference audio is represented as a 2-second MEL-spectrogram, encoder into a sequence of acoustic features per frame of 0.2 seconds duration with the shape of $\mathbb{R}^{5609 \times 512}$.

Our neural model's newly inducted input text prompt undergoes Byte-Pair Encoding (BPE) and Tokenization (Zouhar et al., 2024) to convert textual information into a feature vector $\mathbf{f}_t \in \mathbb{R}^{512.T}$. This feature vector enables context-specific animations, allowing the synthesized video to align with the intended spoken words and expressions implied in the text. The purpose of concatenating $\mathbf{f}_t$ with the combined feature of reference audio $\mathbf{f}_a \oplus \mathbf{f}_s$ is to obtain the speaker's signature in the final flattened feature tokens of $\mathbf{f}_t \oplus \mathbf{f}_a \oplus \mathbf{f}_s \in \mathbb{R}^{5609+T \times 512}$.

Next, we process the input source image through a Variational Auto-Encoder (VAE) (Kingma & Welling, 2022) and a Landmarks Detection model (Zhang et al., 2020). The VAE generates an image embedding $\mathbf{f}_i$, representing the visual style and identity of the person in the source image. Concurrently, the landmarks detection network extracts structural features – face mask feature $\mathbf{f}_{fm}$ and lip mask feature $\mathbf{f}_{lm}$, which are combined with the image embedding vectors to create a fused visual feature representation $\mathbf{f}_i \oplus \mathbf{f}_{lm} \oplus \mathbf{f}_{fm} \in \mathbb{R}^{3136 \times 512}$. The straightforward tendency of traditional methods is either to introduce prior 3D morphable models faces (Zhang et al., 2023b), motion priors of the facial parts (Jang et al., 2024), or guiding video frames (Wang et al., 2022) to learn nuances of facial articulation in relation to the audio in combined latent space. In contrast, we show that the entanglement of multiple latent spaces of text-audio-video using Transformer encoders (Vaswani et al., 2023) can eliminate the dependency on strong motion priors. As a result, we are able to use text prompt features as a set of anchoring tokens to both the Transformer encoders.

### 3.2 ENTANGLEMENT OF COMBINED TEXT-AUDIO-VIDEO LATENT SPACE.

As illustrated in Figure. 2, a smooth synergy between the text-audio latent embedding and the text-image latent embedding is established by two Transformer encoders followed by latent diffusion-guided (Xu et al., 2024b) synthesizer of visual nuances and decoder-only GPT-2 (Casanova et al., 2024) model for synthesizing text-conditioned audio latent.

The first Transformer encoder spatially contextualizes the audio MEL-spectrogram tokens using a dual-stream cross-modal attention mechanism with the flattened version, denoted by $\mathbf{L}(.)$, of *categorically fixed speaker* embedding tokens merged with varying text embedding tokens, *i.e.,* $\mathbf{Q}_a = \mathbf{L}(\mathbf{f}_a \oplus \mathbf{f}_s)$, as

$$\text{Cross-Attention}(\mathbf{Q}_a, \mathbf{K}_{ti}, \mathbf{V}_{ti}) = \text{SoftMax}\left(\frac{\mathbf{Q}_a \mathbf{K}_{ti}^\top}{\sqrt{d_k}}\right)\mathbf{V}_{ti}, \tag{1}$$

where the query vector $\mathbf{Q}_a$ is of dimension $\mathbb{R}^{5609 \times 512}$ and the key-value paring $(\mathbf{K}_{ti}, \mathbf{V}_{ti})$ between the tokens of $\mathbf{L}(\mathbf{f}_t \oplus \mathbf{f}_i \oplus \mathbf{f}_{lm} \oplus \mathbf{f}_{fm})$ has a variable spatial length (padded up-to a max length)

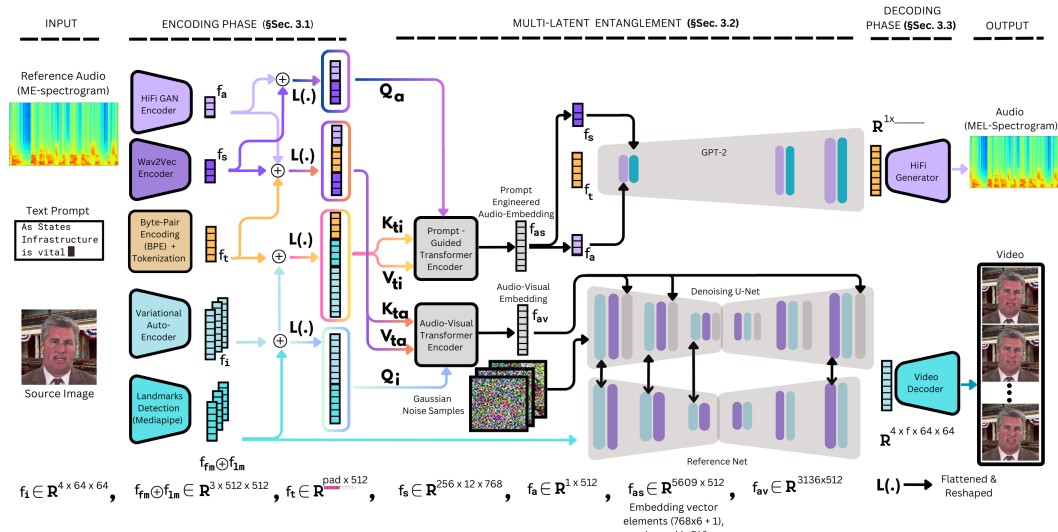

Figure 2: **Our Network Architecture:** Text Prompt-guided joint audio-visual learning representations using dual stream Transformer Encoders and Denoising Diffusion model. The model architecture can be divided into three phases – namely *Encoding Phase*, *Multi-Latent Entanglement*, and *Decoding Phase*. As an output, an audio-visual animation is generated from a single source image, reference audio, and a short text prompt.

with a fixed channel length of 512. Merging the varying text tokens serves two purposes – **(1)** first, querying audio tokens as well as the speaker tokens has been implicitly prompt-engineered by the text tokens, **(2)** second when the resulting prompt-engineered latent embedding vectors $\mathbf{f}_{as}$ are split into its respective constituents, they become proxy weights of text-image embedding vectors.

Similar to the previous encoder block, the second Transformer encoder spatially contextualizes the input masked-image embedding vectors $\mathbf{L}(\mathbf{f}_i \oplus \mathbf{f}_{fm} \oplus \mathbf{f}_{lm})$ using cross-modal attention with the key-value pairs $(\mathbf{K}_{ta}, \mathbf{V}_{ta})$ of merged text-audio embedding tokens $\mathbf{L}(\mathbf{f}_t \oplus \mathbf{f}_a \oplus \mathbf{f}_s)$ similar to the equation 1 as

$$\text{Cross-Attention}(\mathbf{Q}_i, \mathbf{K}_{ta}, \mathbf{V}_{ta}) = \text{SoftMax}\left(\frac{\mathbf{Q}_i \mathbf{K}_{ta}^{\top}}{\sqrt{d_k}}\right) \mathbf{V}_{ta}. \qquad (2)$$

As a result, the output latent embedding on audio-visual features $\mathbf{f}_{av}$ can serve as a compact and compressed representation of facial animation sequences in the high-dimensional space. Therefore, our next step is to learn a synthesizer *i.e.,* a hierarchical latent diffusion model Xu et al. (2024b) for video generation and a corresponding MEL-spectrogram synthesizer based on the X-Text-to-Speech (XTTS) model Casanova et al. (2024).

**Latent Text Conditioned Spectrogram Synthesizer:** The GPT-2 encoder is based on the TTS model (Casanova et al., 2023) and (Shen et al., 2018). This part is composed of a decoder-only transformer module that is conditioned by the audio and speaker embedding vectors $\mathbf{f}_a, \mathbf{f}_s$ disentangled from the prompt-engineered audio embedding vector $\mathbf{f}_{av}$, and the auto-regressive generation of spectrogram tokens is fully driven by the input text tokens from $\mathbf{f}_{av}$.

**Text-Anchored Audio-Video Latent Conditioned Denoising Diffusion:** The Denoising Diffusion model aims to reverse a diffusion process(Ho et al., 2020; Song et al., 2022) that progressively adds random Gaussian noise to data. Inspired by the Hallo method (Xu et al., 2024b), we employ an additional augmentation of the text-anchored latent embedding vector learned to combine the audio and motion nuances on a single image inside the Denoising U-Net (Ronneberger et al., 2015) model of Hallo. The model is initialized with pre-trained weights and fine-tuned during the training step.

Throughout each step of the diffusion process, we introduce embedding cross-attention, which incorporates the combined latent space embedding, particularly our $\mathbf{f}_{av}$, into each diffusion step. This cross-attention mechanism allows the diffusion models to leverage the shared information across modalities, ensuring that the generated outputs (audio and video) are consistent with the input

embedding. The inclusion of cross-attention helps to maintain coherence between the synthesized motion across all the pixels of the source image.

Additionally, diffusion cross-attention facilitates mutual information exchange between the audio and video diffusion blocks. This cross-attention mechanism enables the audio and video models to synchronize their outputs, ensuring that the generated audio and video components are temporally aligned. By integrating this cross-attention, our framework effectively coordinates the diffusion processes, leading to synchronized and coherent multimedia output.

### 3.3 DECODING PHASE FOR AUDIO-VIDEO GENERATION

The outputs of the previous steps are processed by their respective final decoders. For audio generation, similar to the XTTS method (Casanova et al., 2024), the synthesized spectrogram is passed through a Vocoder component of HiFi Generator module to obtain the final audio signal. For video, the Denoising UNet generates $f$ number of frames of dimension $\mathbb{R}^{4 \times f \times 64 \times 64}$, which are decoded by a pre-trained decoder component of (Kingma & Welling, 2019) to produce the complete video.

### 3.4 LOSS FUNCTIONS

To train our model, we use –

**(1)** Video Loss as the Pixel-wise L1 Loss *i.e.,* sum of the $N$ number of pixel intensities between the ground truth image frame $\mathcal{I}_{\text{gt}}^f$ and the generated frame $\mathcal{I}_{\text{gen}}^f$ for all the $f$ number of frames as $\mathcal{L}_{\text{video}} = \sum_f \sum_{i=1}^N \|(\mathcal{I}_{\text{gt}}^f)^i - (\mathcal{I}_{\text{gen}}^f)^i\|$, **(2)** Audio Loss as the Spectrogram MSE loss at the spectrogram $\mathcal{S}$ domain as mean squared error between the ground-truth magnitudes and generated magnitudes at different of time step $t$ as $\mathcal{S}_{\text{gt}}^t$ and the generated frame $\mathcal{S}_{\text{gen}}^t$ as $\mathcal{L}_{\text{audio}} = \frac{1}{T} \sum_{t \in T} \|(\mathcal{I}_{\text{gt}}^f)^i - (\mathcal{I}_{\text{gen}}^f)^i\|^2$. Total loss as $\mathcal{L}_{\text{Total}} = \lambda \mathcal{L}_{\text{audio}} + \mathcal{L}_{\text{video}}$ with balancing factor $\lambda = 0.1$.

## 4 EXPERIMENTAL RESULTS

### 4.1 DATASETS, PREPROCESSING, IMPLEMENTATION DETAILS AND EVALUATION MATRICES

**Datasets:** We have primarily conducted our experiments on 4 datasets. Our model training was done on a combination of **VoxCeleb** Dataset (Nagrani et al., 2019), **FakeAVCeleb** dataset (Khalid et al., 2022), **HDTF** (Zhang et al., 2021) and the **CelebV-HQ** dataset (Zhu et al., 2022). VoxCeleb is an audio-visual dataset consisting of short clips of human speech, extracted from interview videos uploaded to YouTube. FakeAVCeleb is a novel audio-video multimodal deepfake dataset. We only considered the non-deepfake part of the dataset. CelebV-HQ is a large-scale video facial attributes dataset demonstrating a diverse quality of data, which is important to test the robustness of our model. HDTF is a large in-the-wild high resolution audio-visual dataset built for talking face generation.

**Preprocessing:** Our preprocessing involved resizing the videos to 512x512 and then cropping each video sample to the first 20 seconds (at 25FPS which equates to 500 frames). We then separated the audio from the video using ffmpeg, and then ran the OpenAI's Whisper model(Radford et al., 2022) to transcribe the audio speeches.

**Implementation details:** The optimizer used for our model is AdamW with a learning rate of 1e-4 and weight decay of 1e-2, and the scheduler has a step-wise learning rate with a step size of 1000 and gamma of 0.5. The weight decay regularizes the model, preventing any overfitting. We have used Nvidia 1xA6000s GPU for training each model, and the model inference requires 12GB of VRAM. The total parameter size of the model comes to 1,575,936 and performs 5.39 GFLOPs (Giga Floating Point Operations) per generation. We have trained the models for 10 epochs, with a batch size of 8. The Hifi-Gan, Wav2Vec Encoders, the Variational Autoencoder, Diffusion Models, and the GPT2 Decoder are pre-trained, which were further trained with the rest of the entire proposed network.

**Evaluation Metrics:** Following are the evaluation matrices employed.

**Video Metrics**: *Fréchet Video Distance (FVD:* A measure of the quality of generated videos, comparing them to real videos based on spatio-temporal features. Lower values indicate better performance (Unterthiner et al., 2019). *FID (Fréchet Inception Distance):* Evaluates the visual quality of individual frames by comparing the distributions of generated and real images. Lower scores represent better visual quality (Heusel et al., 2018). *Fréchet Video Motion Distance(FVMD):* Measures the quality of motion in generated videos, capturing the difference between real and generated motion trajectories. Lower values signify a more realistic motion.(Liu et al., 2024).

**Audio Metrics**: *Fréchet Audio Distance (FAD):* Assesses the similarity between generated and real audio samples, with lower scores indicating closer resemblance. *Short-Time Objective Intelligibility (STOI ):* Measures the intelligibility of the generated speech. Higher values represent more intelligible speech (Kilgour et al., 2019). *Mel Cepstral Distortion(MCD):* A metric used to evaluate the quality of speech synthesis by comparing the spectral features of generated and reference audio. Lower scores imply better audio quality (Zezario et al., 2020).

**Audio-visual (AV) synchronisation**: We used two metrics proposed in Wav2Lip Prajwal et al. (2020b) to find the audio-visual synchronisation. The first is the average error measure calculated in terms of the distance between the lip and audio representations, "LSE-D" ("Lip Sync Error Distance"). A lower LSE-D denotes a higher audio-visual match, i.e., the speech and lip movements are in synchronization. The second metric is the average confidence score, "LSE-C" (Lip Sync Error Confidence). The higher the confidence, the better the audio-video correlation.

**Training and Testing:** Our primary training dataset is the VoxCeleb dataset(Nagrani et al., 2019), where our training set comprised of approximately 36000 videos. We chose this training set by filtering out individuals whose speech was in English. We tested on more than 200 samples from each of the four datasets (VoxCeleb, FakeAVCeleb, CelebV-Hq and HDTF.), resulting in a test set of over 800 unseen samples.

We benchmarked the video outputs for the unseen samples against SoTA Portrait Animation models, like Hallo(Xu et al., 2024b), Sadtalker(Zhang et al., 2023b), EAT(Gan et al., 2023) and Audio2Head(Wang et al., 2021). We also benchmarked the audio outputs for the unseen samples against SoTA Speech generation models, like Tortoise(Betker, 2022), Your_TTS(Casanova et al., 2023), XTTS_v2(Casanova et al., 2024) and GlowTTS(Kim et al., 2020).

## 4.2 RESULT ANALYSIS

**Video Results**: From Table 1, we can observe that our model shows superior performance across all three metrics FID, FVD, and FVMD on VoxCeleb, CelebV-Hq and HDTF. This indicates high fidelity and minimal discrepancies are attended by the proposed model. On the FakeAVCeleb, the performance is slightly poorer but can be comparable, it still maintains strong visual consistency and realism on visual inspection. For the CelebV-HQ our model excels again, demonstrating its capability to produce high-quality video outputs. On HDTF our model shows incredible performance in the FID and FVD metrics, beating all the other models, while our model is admirably performing considering FVMD when compared to Hallo.

Table 1: Video pipeline evaluation scores across datasets.

| Dataset | Model | FID Score (↓) | FVD Score (↓) | FVMD Value (↓) |
|---|---|---|---|---|
| **VoxCeleb** | Audio2Head | 81.00 | 90.12 | 5100.92 |
| | Hallo | 67.28 | 70.69 | 5703.44 |
| | EAT | 85.16 | 80.38 | 4878.36 |
| | SadTalker | 119.36 | 112.77 | 6352.19 |
| | Our Model | **42.88** | **49.78** | **4192.07** |
| **FakeAVCeleb** | Audio2Head | 93.59 | 97.85 | 1329.23 |
| | Hallo | **26.88** | **39.42** | 2351.20 |
| | EAT | 94.34 | 98.49 | **1324.91** |
| | SadTalker | 81.77 | 77.10 | 4158.18 |
| | Our Model | 47.24 | 49.15 | 2263.54 |
| **CelebV-HQ** | Audio2Head | 90.22 | 102.76 | 2939.49 |
| | Hallo | 42.76 | 56.10 | 2816.68 |
| | EAT | 47.88 | 56.21 | 2894.31 |
| | SadTalker | 52.60 | 52.55 | 2789.19 |
| | Our Model | **34.01** | **43.67** | **2743.29** |
| **HDTF** | Audio2Head | 37.78 | 32.69 | 2633.04 |
| | Hallo | 20.54 | 25.81 | **1290.57** |
| | EAT | 29.57 | 29.34 | 2573.05 |
| | SadTalker | 22.34 | 23.57 | 2410.89 |
| | Our Model | **11.72** | **15.58** | 1784.16 |

Table 2: Audio pipeline evaluation scores across datasets.

| Dataset | Model | FAD Score (↓) | MCD Score (↓) | STOI Score (↑) |
|---|---|---|---|---|
| **VoxCeleb** | Tortoise | 258.54 | 82.37 | 0.10 |
| | Your_TTS | **199.52** | 111.79 | **0.19** |
| | XTTS_v2 | 249.17 | 100.80 | 0.13 |
| | GlowTTS | 329.21 | 103.94 | 0.15 |
| | Our Model | 241.75 | **75.39** | 0.17 |
| **FakeAVCeleb** | Tortoise | 871.14 | 82.12 | 0.10 |
| | Your_TTS | 445.38 | 65.60 | **0.21** |
| | XTTS_v2 | 184.39 | 77.88 | 0.11 |
| | GlowTTS | 482.04 | 87.11 | 0.18 |
| | Our Model | **171.52** | **55.12** | 0.19 |
| **CelebV-HQ** | Tortoise | 529.06 | 113.18 | 0.09 |
| | Your_TTS | 520.01 | 137.58 | 0.16 |
| | XTTS_v2 | 509.90 | 124.61 | 0.07 |
| | GlowTTS | 549.18 | 139.81 | **0.22** |
| | Our Model | **244.83** | **85.76** | 0.18 |
| **HDTF** | Tortoise | 425.30 | 67.15 | 0.11 |
| | Your_TTS | 467.42 | 49.38 | 0.15 |
| | XTTS_v2 | 135.11 | 49.65 | 0.14 |
| | GlowTTS | 510.61 | 66.42 | 0.12 |
| | Our Model | **106.43** | **44.05** | **0.15** |

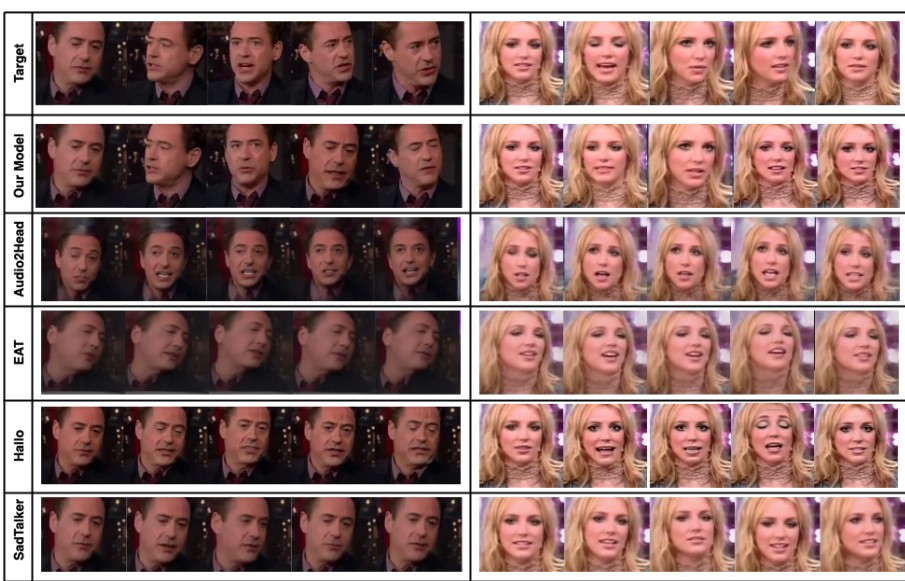

Figure 3: The figures in each row show frames from the videos generated by each technique in the order: Ground Truth, Our proposed Model, Audio2Head (Wang et al., 2021), EAT (Gan et al., 2023), Hallo (Xu et al., 2024b), and SadTalker (Zhang et al., 2023b) on the VoxCeleb Dataset. A frame in each column for both videos corresponds to the same time-stamp (frames were sampled at equal intervals of 25 seconds across the videos).

Based on the results, we observed that for some datasets certain models work slightly better than the proposed model, and the reason behind this is that those models try to memorize certain properties from individual datasets. Whereas our model is a more generalized version that can performed consistently on cross datasets having varying resolution, and video quality. The visualization from Figure 3 also concludes that our model can generate video very close to the ground truth and better than any model. From Figure 4 it can be concluded that our model can generate nearby results for HDTF, FakeAVCeleb and CelbV-HQ when compared to ground truth.



Figure 5: Ground Truth vs. Generated Audio Spectrograms for (a) VoxCeleb, (b) CelebV-HQ, (c) FakeAVCeleb and (d) HDTF datasets

**Audio Results:** We can infer from Table 2 that our model consistently performs the best in the MCD Score metric, which suggests that it minimizes distortion between the spectral features of synthetic and reference speech. While considering the FAD scores, our model also performed on par state-of-the-art, except on VoxCeleb where Your_TTS is better, these showcase that the proposed model can generate consistently similar audio compared to the ground truth. Considering the STOI metric, the performance of our model is similar to or slightly lower than Your_TTS. The analysis of all the measures showcases that our model is more generalized and realistic as it can minimize distortion and also generate accurate distributions, and maintain intelligibility of the speech consistently better than any other models. The visualization from Figure 5 also concludes that our model can generate audio very close to the ground truth.

**AV synchronization results:** From Table 3 we can conclude that our proposed model has performed better audio-video synchronization than SOTA and is close to the ground truth. The proposed model has the lowest LSE-D, i.e. better audio-visual match, i.e. and LSE-C i.e. better audio-video correlation. We have also analyzed the model with varying accents, blurred audio profiles, and audio profiles of a kid with a source image of an adult and vice versa, and the results were found to be effective, no bias was found in

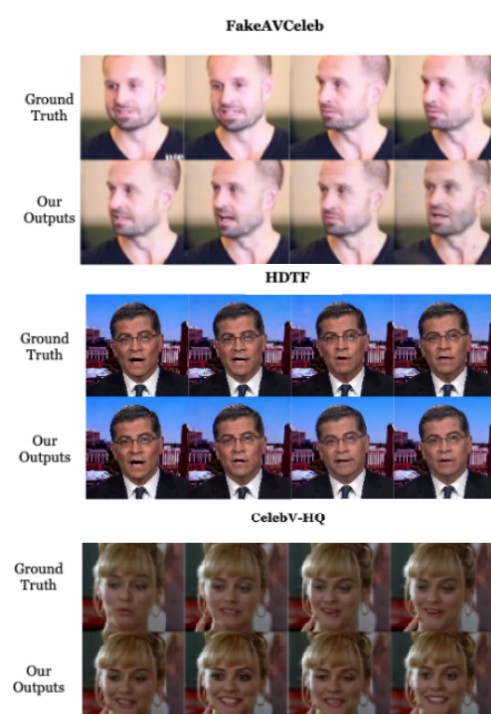

Figure 4: Results of our model on FakeAVCeleb, Celeb-HQ and HDTF datasets.

any aspect. Models fail in a few scenarios where a very noisy audio profile is used, output audio is feeble or for source images with closed eyes face dynamics get affected (details are in supplementary.

### 4.3 ABLATION STUDY

Table 4 shows the ablation study of our proposed model. We have 3 main sub-networks that define the output of our model. The **Transformer Encoder Block(TE)** (Vaswani et al., 2023) with two variations shared-TE **(STE)** where both audio and video pipeline shares a transformer block and explicit-TE **(ETE)** where audio and video pipeline has explicit or separate transformer block. **Diffusion(Song et al., 2022) Cross Attention(DC)**, and the **Embedding Cross Attention(EC)**. From our results, it is understandably explainable that the transformer encoder block, which encodes our in-

Table 3: Evaluation of audio-visual synchronization

|  | LSE-C(↑) | LSE-D(↓) |
|---|---|---|
| Groundtruth | 5.45 | 8.52 |
| Hallo | 3.03 | 8.71 |
| Audio2Head | 2.51 | 10.34 |
| EAT | 4.39 | 9.35 |
| SadTalkert | 5.44 | 10.09 |
| STE | 5.71 | 8.41 |
| ETE/ Proposed | 5.74 | 8.38 |

puts into a common latent space, is the most important modality of our network, with its removal drastically reducing our metric values. Our experiments also show that the cross-attention blocks between the diffusion models are more important than the embedding cross-attention since our metric values drop more when we remove the diffusion cross-attention, probably since the diffusion cross-attention already syncs the modalities during the parallel learning stage. Another important aspect of ablation is the encoding latent in the individual transformer i.e. ETE is much better than STE. This infers that it is important to encode the latent for each modality separately while sharing information among the generated modalities. Table 5 shows our ablation study on the encoders. "Only Visual Tokens Attended" involves eliminating the audio prompt-guided transformer. Similarly, the "Only Audio Tokens Attended" involves using only the audio prompt-guided transformer. "No Hifi-GAN" and "No Wav2Vec" are results obtained by eliminating the encoding process of the Hifi-GAN and Wav2Vec Models respectively. "No Visual token in prompt guided-Transformer" involves not attending the visual tokens in the prompt guided-Transformer. These ablations quantify the importance of each of the components.

Table 4: Ablation study of the transformers.

| ETE | STE | DC | EC | FID ($\downarrow$) | FVD ($\downarrow$) | FVMD ($\downarrow$) | FAD Score ($\downarrow$) | MCD ($\downarrow$) | STOI ($\uparrow$) |
|---|---|---|---|---|---|---|---|---|---|
| | | ✓ | ✓ | 86.70 | 80.88 | 5275.89 | 328.27 | 95.44 | 0.07 |
| | ✓ | | | 68.83 | 74.19 | 4412.74 | 260.91 | 87.51 | 0.11 |
| | ✓ | ✓ | | 63.68 | 71.38 | 4298.30 | 250.12 | 83.96 | 0.14 |
| | ✓ | ✓ | ✓ | 61.44 | 69.15 | 2720.41 | 241.77 | 81.60 | 0.17 |
| ✓ | | ✓ | ✓ | **42.88** | **49.78** | **4192.07** | **241.75** | **75.39** | **0.17** |

Table 5: Ablation study of the encoders.

| Ablation | FID ($\downarrow$) | FVD ($\downarrow$) | FVMD ($\downarrow$) | FAD Score ($\downarrow$) | MCD ($\downarrow$) | STOI ($\uparrow$) |
|---|---|---|---|---|---|---|
| Only Visual Tokens Attended | 68.31 | 78.42 | 5747.04 | 304.98 | 81.17 | 0.13 |
| Only Audio Tokens Attended | 69.02 | 79.35 | 6576.85 | 301.49 | 80.65 | 0.13 |
| No Hifi-GAN | 85.25 | 94.28 | 7483.40 | 498.33 | 87.51 | 0.09 |
| No Wav2Vec | 70.10 | 80.96 | 5926.64 | 309.95 | 89.58 | 0.11 |
| No Visual token in Prompt Guided transformer | 54.38 | 62.02 | 5481.36 | **221.07** | **63.25** | 0.12 |
| Proposed Model | **42.88** | **49.78** | **4192.07** | 241.75 | 75.39 | **0.17** |

## 4.4 SOCIAL RISKS AND MITIGATIONS

There are social risks with technology development for text-driven audio video talking face generation. The foremost risk is the ethical implications of creating highly realistic talking faces, it can be used for malicious purposes, such as deepfakes. To mitigate such risk, ethical guidance for the use of such generation techniques is required. Also, concerns regarding privacy and consent are implicit in such work. Transparent data usage policies by consent, and safeguarding the privacy of individuals can mitigate such concerns. By addressing these we aim to promote responsible and produce ethical generative technology.

## 5 CONCLUSION

This paper introduces a novel method for realistic speaking and talking faces by joint multimodal video and audio generation. We provide a holistic architecture where the information is exchanged between the modalities via the proposed multi-entangled latent space. A source image of an individual as a driving frame, reference audio which can be referred to as the audio profile of the individual and a driving or prompt text is passed as an input. The model encodes the input driving image, prompt/driving text, and the voice profile which are further combined and passed to the proposed multi-entangled latent space consisting of two separate transformers and diffusion block for video and text decoder for audio pipeline to foster key-vale and query representation for each modality. By this spatiotemporal person-specific featuring between the modalities is also established. The entangled-based learning representation is further passed to the respective decoder of audio and video modality for respective outputs. Conducted experiments and ablation studies prove that the proposed multi-entangled latent-based learning representation has helped our model obtain superior results on both video and audio outputs as compared to state-of-the-art models. While there is always scope for improvement in the future, we believe that our model has shown promising new learning representation for realistic speaking and talking face generation models.

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
