# Supplementary for ICLR 2025: Submission ID 10887

This document provides supplementary materials omitted from the main paper owing to space constraints. In Section 1 details the model architecture. In Section 2 we describe the analysis of models with respect to STE and ETE and a details analysis of the ablation study, we included additional visualizations in Section 3. Section 4 consists of a discussion on limitations.

## 1 DETAILS OF MODEL ARCHITECTURE

The following table summarises the parameters of our transformer encoder. Both the Transformer Encoders have the same architecture just the input shapes are different. For the Shared Transformer Encoder Model, the number of parameters hence is half the number of parameters of the Explicit Transformer Encoder.

Table 1: Layer details and parameters

| Layer (type) | Output Shape | Param # | Tr. Param # |
|---|---|---:|---:|
| LayerNorm-1 | [3136, 512] | 1,024 | 1,024 |
| MultiheadAttention-2 | [3136, 512], [3136, 5610] | 1,048,576 | 1,048,576 |
| LayerNorm-3 | [3136, 512] | 1,024 | 1,024 |
| Linear-4 | [3136, 512] | 262,144 | 262,144 |
| LayerNorm-5 | [3136, 512] | 1,024 | 1,024 |
| Linear-6 | [3136, 512] | 262,144 | 262,144 |
| **Total params:** | | **1,575,936** | **1,575,936** |

## 2 ABLATION STUDY ON STE AND ETE

Table 2 represents a comparison of two versions of our model, one with a shared transformer encoder (STE) and the other separately (ETE). The former uses the same weights in both the video and audio transformers for the respective query, key, and value matrices, while also sharing the same weights for the fully connected layer. The latter has separate query, key, and value matrices for the audio and video parts, thus allowing for controlled independence in learning the audio and video representations. The Shared Weights model also enforces the sharing weights of the subsequent fully connected layers, enforcing a reduction in the number of learning parameters.

Evidently, in almost all of the parameters the Explicit Transformer Encoder Model(ETE), which is our final proposal, performs better than the Shared Transformer Encoder Model(STE), as it allows for deeper learning representations of the Audio and Video Pipeline, while still maintaining a certain coherence between the generations.

The primary goal of the model is to generate the audio as per the input audio profile and the video as per the source image along with the audio-visual synchronization. Hence, the model aims to learn the personal characteristics that are provided as inputs via the source image and reference audio profile. The cross-attention mechanism enables the audio and video models to synchronize their outputs, ensuring that the generated audio and video components are temporally aligned.

To ensure that the proposed cross-attention does not add a bias to audio or video generation, specific feature engineering by multi-latent entanglement is performed. As we can see in Fig 2, the encoded features from the prompt text and audio samples i.e. the output of the word2vec, HiFiGAN encoder and BPE are passed along with the cross attention from

Table 2: Ablation results on STE and ETE for different datasets.

| Dataset | Weights | FAD ($\downarrow$) | MCD ($\downarrow$) | STOI ($\uparrow$) | FID ($\downarrow$) | FVD ($\downarrow$) | FVMD ($\downarrow$) |
|---|---|---|---|---|---|---|---|
| **VoxCeleb** | Shared TE | 193.42 | 67.06 | 0.124 | 34.21 | 39.97 | **2720.41** |
| | Explicit TE | **241.75** | **75.39** | **0.17** | **42.88** | **49.78** | 4192.07 |
| **FakeAVCeleb** | Shared TE | 174.68 | 56.47 | 0.10 | 47.25 | 49.65 | 2284.69 |
| | Explicit TE | **171.52** | **55.12** | **0.19** | **47.24** | **49.15** | **2263.54** |
| **CelebV-HQ** | Shared TE | 246.53 | 87.29 | 0.11 | 35.05 | 44.30 | **2658.90** |
| | Explicit TE | **244.83** | **85.76** | **0.18** | **34.01** | **43.67** | 2743.29 |
| **HDTF** | Shared TE | 108.36 | 49.72 | 0.12 | **11.56** | 15.94 | 1785.62 |
| | Explicit TE | **106.43** | **44.05** | **0.15** | 11.72 | **15.58** | **1784.16** |

prompt guided transfer to the GPT2 decoder and then to HiFi generator to generate the voice profile specific audio output. Similarly, for video pipeline output latent embedding on audio-visual features along with the visual tokens is passed to the reference net that can serve as a compact and compressed representation of facial animation sequences in the high-dimensional space which can be further decoded to get the source-image specific video.

From the ablation, we can also find that biases are handled by the proposed model, Shared transformer encoders (STE) perform worse than the explicit transformer encoders (ETE). Also, ablation with no video tokens in the Prompt Guided transformer the temporal comprehensiveness of the audio and synchronization matrix is worse. We can also observe that the synchronization matrix gets worse w/o attention.

## 3 ADDITIONAL QUALITATIVE OUTPUTS

The following section presents additional outputs that prove our model's versatility and quality of data generation. Figures 1 show that we are able to generate very high-definition output videos, provided that the input driving frame is a high-resolution image. Our model maintains the resolution and doesn't add noise to the image over the duration of the video, as evidenced by the outputs: Frame-1, Frame-25, Frame-50, and Frame-100 of our generated video.

## 4 LIMITATIONS

### 4.1 DRIVING IMAGE

While we have been able to solve issues that most video-generative models face, like lip-sync, and audio-video coherence, there are still some problems to tackle.

### 4.2 INPUT AUDIO PROFILE

We have observed that any case of corrupt input audio profile(feeble, unclear, or noisy) results in an output with very little audio. While the frames are generated correctly still, the model requires an adequately intelligible audio profile input.

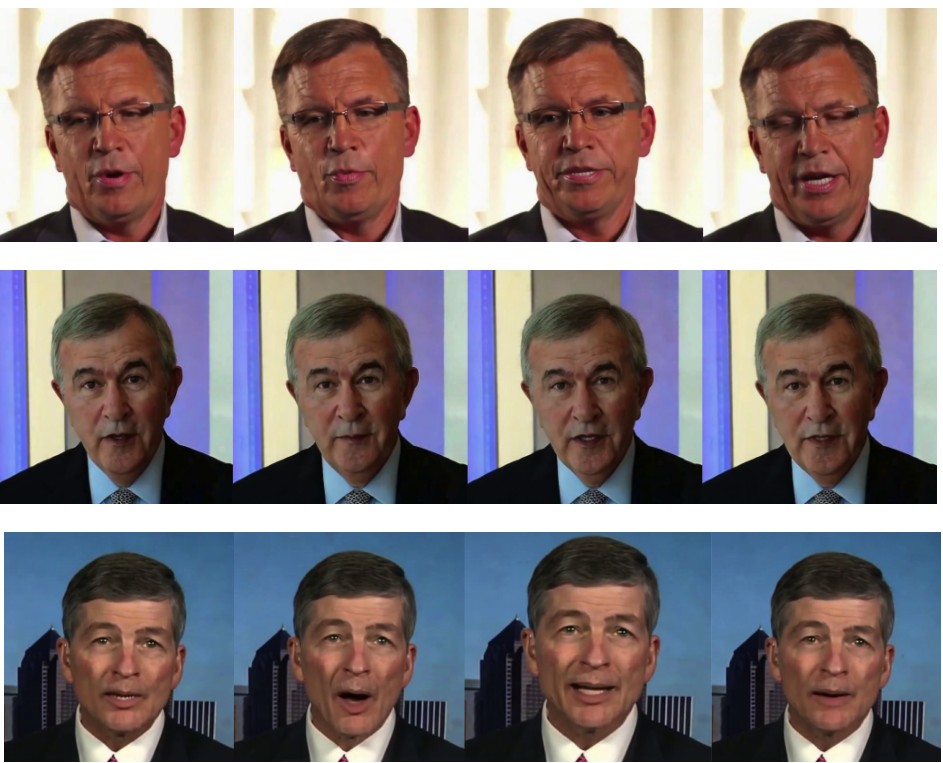

Figure 1: The outputs of high definition quality.

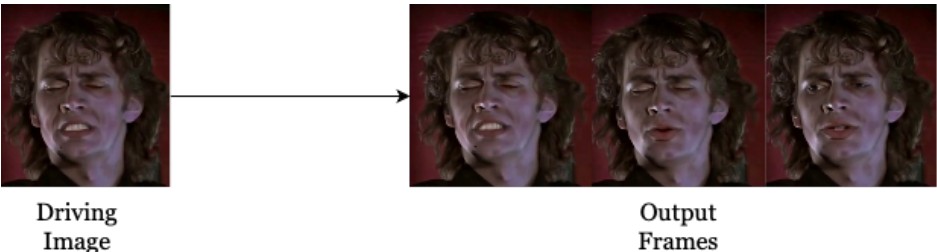

Figure 2: Driving Images like this example with closed eyes result in a lot of generated frames having closed eyes, for it is difficult to assume the eye characteristics for a particular source image.

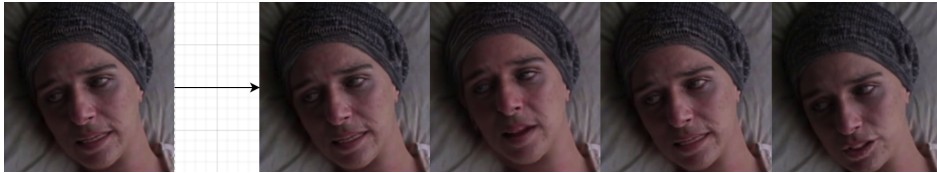

Figure 3: In rare cases of generated videos, the eyes always point to a single direction, which is slightly un-human-like. Although this is rare, it can be pointed out as a limitation to the model in a few cases which doesn't involve the driving image subject posing directly at the camera.

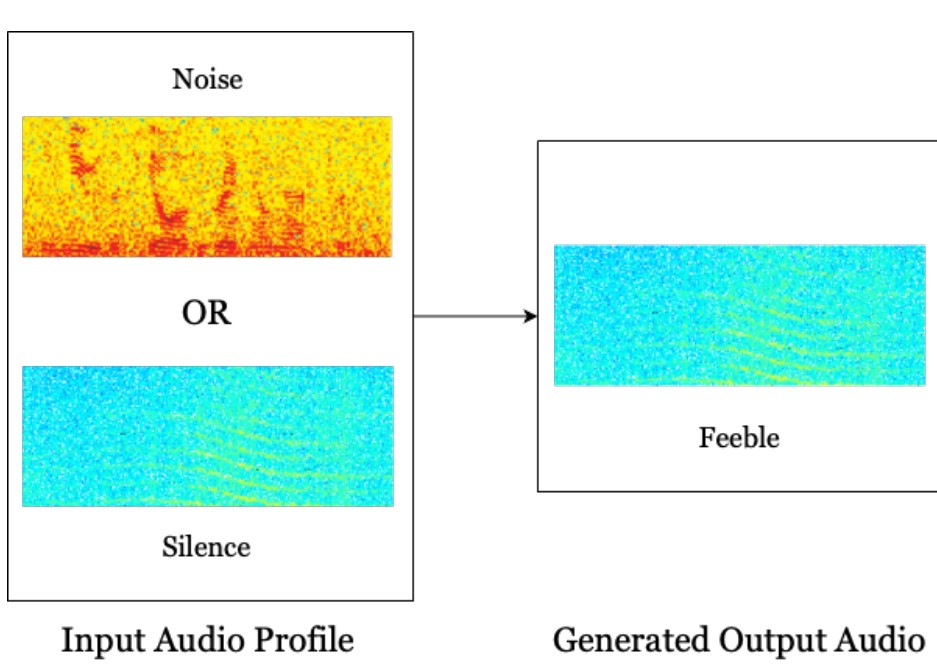

Figure 4: Feeble output audio generation when input audio profile is very noisy or blank.