# OpenReview forum: "Playing For You: Text Prompt-guided Joint Audio-visual Generation for Narrating Faces using Multi-entangled Latent Space"
_ICLR.cc/2025/Conference — Submitted to ICLR 2025_

### Official Review · Reviewer_RNFb · 2024-10-29

**Soundness:** 3
**Presentation:** 3
**Contribution:** 3
**Rating:** 6
**Confidence:** 4

**Summary:**

The paper introduces a multimodal model aimed at generating synchronized audio-visual output based on input prompts. It combines static images, voice profiles, and target texts in a multi-entangled latent space to produce both speech and corresponding facial movements. The model comprises three main phases: encoding, entangled latent space formation, and decoding for generating output audio and video

**Strengths:**

- Introducing a multi-entangled latent space for synchronizing audio and visual modalities, potentially bridging the gap between current single-modality generation models and real-world applications requiring synchronized outputs.

- The model was ested on multiple datasets (VoxCeleb, CelebV-HQ, HDTF).

**Weaknesses:**

- Utilizing HiFi-GAN, Wav2Vec, and multiple Transformers implies significant computational resources, but the paper fails to discuss resource efficiency, scalability, or training costs. This may pose practical limitations, especially in deployment on lower-resource devices.

- The "multi-entangled latent space" concept, while promising, lacks clarity in its practical execution, especially regarding how it manages and aligns the complex temporal features across audio and visual modalities. Detailed insight into the entanglement mechanism would enhance the model's comprehensibility and reproducibility.

- While lip synchronization is briefly mentioned, it lacks thorough quantitative and qualitative analysis. Lip-sync accuracy and expressiveness are critical in applications requiring realistic facial animations, yet these elements are not comprehensively evaluated in the paper.

- Given the potential misuse in generating realistic, personalized talking faces, ethical considerations are necessary. The authors should address potential implications, such as privacy risks, unauthorized identity mimicry, or the generation of deepfakes, to ensure responsible use.

**Questions:**

- Can you clarify the role and specifics of the multi-entangled latent space? How does it manage spatiotemporal alignment across the modalities?

- Are there ethical safeguards or limitations implemented to mitigate misuse, given the realistic nature of the generated outputs?

- Missing important previous work

Xu, Chao, et al. "FaceChain-ImagineID: Freely Crafting High-Fidelity Diverse Talking Faces from Disentangled Audio.",CVPR 2024

**Details Of Ethics Concerns:**

Privacy, security, and potential misuse in creating realistic yet unauthorized identity mimicry. Further discussion on safeguards or ethical limitations would be advisable.

---

> ### Author Response · Authors · 2024-11-21
> **Author response**
>
> We sincerely thank the reviewer for providing helpful feedback on our work. Based on that feedback, we will add the following to our revised paper:
>
> C1: Utilizing HiFi-GAN, Wav2Vec, and multiple Transformers implies significant computational resources, but the paper fails to discuss resource efficiency, scalability, or training costs. This may pose practical limitations, especially in deployment on lower-resource devices.
>
> --Re: The parameter details for the transformer used are discussed in the supplementary. The total parameter size of the model is 1,575,936 M. The model does 5.39 GFLOPs for the whole generation pipeline. The generation time for a 10-second is around 2 seconds with 12GB of VRAM, hence we believe it can be applied for real-time generation.
>
>
> C2: The "multi-entangled latent space" concept, while promising, lacks clarity in its practical execution, especially regarding how it manages and aligns the complex temporal features across audio and visual modalities. Detailed insight into the entanglement mechanism would enhance the model's comprehensibility and reproducibility.
>
> --Re: Apology for the confusion as we have mentioned in the introduction section of the original paper (122-133) and the proposed methodology (section 3.2).  To summarize, the multi-entangled latent space integrates embeddings from three modalities—text (CLVP encoded), audio (autoregressive encoded), and image (VAE encoded). This unified latent representation is created using transformer encoders that perform cross-attention and self-attention mechanisms to capture the interdependencies between the modalities (both spatial and temporal synergy).
>
> C3: While lip synchronization is briefly mentioned, it lacks thorough quantitative and qualitative analysis. Lip-sync accuracy and expressiveness are critical in applications requiring realistic facial animations, yet these elements are not comprehensively evaluated in the paper.
>
> --Re: Thank you for your valuable insight. From the literature, it can be found that lip synchronization is a good means for evaluation of audio-visual synchronization (along with critical aspects like coherence, smoothness, synchronization, realisticness and expressiveness). We did experimentation with two metrics as mentioned in Wav2Lip [1]. The first is the average error measure calculated in terms of the distance between the lip and audio representations, “LSE-D" (“Lip Sync Error - Distance"). A lower LSE-D denotes a higher audio-visual match, i.e., the speech and lip movements are in sync. The second metric is the average confidence score, “LSE-C" (Lip Sync Error - Confidence). The higher the confidence, the better the audio-video correlation. A lower confidence score denotes that there are several portions of the video with completely out-of-sync lip movements. From the below table, we can conclude that our proposed model has performed better than the other SOTA models and is close to the ground truth because we generate both modalities.
>
> 1: Prajwal, K R et al.,. A Lip Sync Expert Is All You Need for Speech to Lip Generation In the Wild. Proceedings of the 28th ACM International Conference on Multimedia
>
>
>
>                                       LSE-C(↑)         LSE-D(↓)
>      Groundtruth                      5.45             8.52
>      Combined Encoder                 5.71             8.41
>      Hallo                            3.03             8.71
>      Audio2Head                       2.51            10.34
>      EAT                              4.39             9.35
>      SadTalker                        5.44            10.09
>      Proposed                         5.74             8.38
>
>
> C3: Given the potential misuse in generating realistic, ......
>
> --Re: Thanks for pointing this out, we will address the potential ethical risk and possible misuse in the revised version.
>
> C4: Questions:
> Can you clarify the role and specifics of the multi-entangled latent space? How does it manage spatiotemporal alignment across the modalities?
>
> --Re: For spatiotemporal alignment, the latent space ensures consistent mapping across modalities by incorporating temporal features from the audio and spatial features from the image. The cross-attention mechanism dynamically aligns these features during each step of the diffusion process, maintaining coherence in speech and animation while preserving the reference voice and image details.
>
> C5: Are there ethical safeguards or limitations implemented to mitigate misuse, given the realistic nature of the generated outputs?
>
> --Re: Thank you for your valuable feedback. Considering the nature of the Deepfake generation, this is an important point. We will be adding some guard rails to our model (such as filtering the prompt and the final generation)when we release our inference API on the hugging face.
>
> C6: Missing important previous work
> Xu, Chao, et al. "FaceChain-ImagineID: ....
>
> --Re: Thanks for pointing this out, we will carefully add the previous works in the revised version.

---

### Official Review · Reviewer_4YGG · 2024-11-03

**Soundness:** 2
**Presentation:** 2
**Contribution:** 1
**Rating:** 3
**Confidence:** 3

**Summary:**

This paper addresses the combined tasks of text-to-speech modeling, which generates audio from text input, and talking face modeling, which produces a video of a person speaking for audio or text as input. In contrast to previous work (Faces that Speak by Jang et al., 2024) that solved this combined task (inputs: text, image. outputs: audio, talking video), this paper incorporates reference audio as an additional input.  That is, the task can be summarized as generating text-to-speech and talking face video with additional conditioning on reference audio (inputs: text, image, reference audio. outputs: audio, talking video)

**Strengths:**

- This approach has potential applications in real-world scenarios.
- The experiments demonstrate a measurable performance improvement.

**Weaknesses:**

- The technical contributions in the paper are minimal, as the task is primarily addressed by combining existing architectures trained for individual tasks (e.g., Hallo: Hierarchical Audio-Driven Visual Synthesis for Portrait Image Animation Xu et al. (2024a) for video generation, the MEL-spectrogram synthesizer based on the X-Text-to-Speech (XTTS) model by Casanova et al. (2024), and other various input encoders). Given the reliance on these pre-trained networks from prior works, one might question whether the proposed task could have been approached as a zero-shot solution.

- For the proposed architectural components, there is a lack of ablation studies to demonstrate their relevance, particularly concerning the transformer encoders that facilitate interactions between different modalities before proceeding to the generation modules. It would have been beneficial for the authors to include ablation studies that clarify which modality interactions are essential for each generation task and how performance varies as a result. Currently, the ablation study only provides results for removing both encoders and for sharing encoders.

**Questions:**

- In the prompt-guided transformer encoder attention, how would removing the vision tokens impact audio generation? What role do the vision tokens play in audio generation?
- In Tables 1 and 2, could you clarify which reported performance values of previous works were re-evaluated by you and which were directly taken from the original papers? It seems that many values have been re-evaluated. If all the reported values are re-evaluated, why were at least some commonly used benchmark datasets not employed to allow for direct comparisons with other papers?
- Does using reference audio during training cause the model to develop biases that link specific visual facial features with certain audio profiles? At inference time, can the model overcome these potential biases? For example, if a child’s voice is used as the reference audio but the input image is of an adult male, will the generated voice adapt to match the adult’s appearance, or retain the characteristics of the child’s voice?
Note: If the output does not retain the childlike qualities of the reference audio, this indicates the model has learned a bias.

---

> ### Author Response · Authors · 2024-11-21
> **Author response**
>
> We sincerely thank the reviewer following is our response:
>
> C1: The technical contributions in the paper are minimal,...
>
> ---Re: Apology for the confusion. We have not used any existing architecture rather we used some of the components from the SOTA such as Hifi-Gan, Wav2Vec Encoders, Variational Autoencoder, Diffusion Models, and the GPT2 Decoder. The multi-entangled latent space integrates embeddings from three modalities—text (CLVP encoded), audio (autoregressive encoded), and image (VAE encoded) and further after cross and self-attention to individual decoders of the modalities. This unified latent representation is created using transformer encoders that perform cross-attention and self-attention mechanisms to capture the interdependencies between the modalities (both spatial and temporal synergy).
>
> All of the metrics were re-evaluated. This was because we only used the videos of English speakers from all the datasets which we could transcribe using the Whisper model. We also trimmed the duration of videos to 10 seconds during preprocessing which is needed for our model.
>
> C2: For the proposed architectural components, there is a lack of ablation ...
>
> ----Re: Thanks for your insight. We have done additional ablation suppressing the encoders and latent entanglements. Fowwling are the results
>
>                             FID (↓)           FVD (↓)          FVMD (↓)           FAD Score (↓)       MCD (↓)          STOI (↑)
>     Video Transformer only     68.31	   78.42	   5747.04	        304.98	           81.17	    0.13
>     Audio Transformer only     69.02	   79.35	   6576.85	        301.49	           80.65	    0.13
>     No Hifigan                 85.25	   94.28	    7483.4	        498.33 	           87.51	    0.09
>     No wav2vec                 70.1	           80.96	   5926.64 	         309.95	           89.58	    0.11
>     Proposed                  42.88            49.78            4192.07              241.75            75.39            0.17
>
> C3:  removing the vision tokens impact audio generation?.....
>
> ---Re: From the above Table, we can conclude that having Attention Mechanisms for only 1 modality adversely affects the performance for the other modality. The visual embeddings used as cross-attention in the prompt-guided transformer encoder is capturing some complex spatiotemporal relationship, e.g. a complex combination of features like the person's facial features and expressions.
>
> C4: In Tables 1 and 2, could you clarify which reported performance ..
>
> --Re: All of the metrics were re-evaluated(as mentioned above).These were the datasets used by the previous works. Sadtalker uses preprocessed HDTF and VoxCeleb. Audio2Head uses VoxCeleb, GRID, and LRW. Hallo uses CelebV and HDTF. EAT uses Celeb2 and MEAD. We tried to use most of the common datasets used by the previous works. This are experiments on zero short while AV is learned jointly
>
> C5: develop biases that link specific visual facial features ...?
>
> --Re: Thanks for your interesting question. To investigate the impact of the mentioned bias we have evaluated the model with lip synchronization Wav2Lip [1]. The first is the average error measure calculated in terms of the distance between the lip and audio representations, “LSE-D" (“Lip Sync Error - Distance"). A lower LSE-D denotes a higher audio-visual match, i.e., the speech and lip movements are in sync. The second metric is the average confidence score, “LSE-C" (Lip Sync Error - Confidence). The higher the confidence, the better the audio-video correlation. A lower confidence score denotes that there are several portions of the video with completely out-of-sync lip movements. Results show better performance of the proposed method.
>
> 1: Prajwal, K R et al.. A Lip Sync Expert Is All You Need for Speech to Lip Generation In the Wild. Proceedings of the 28th ACM MM
>
>
>                                       LSE-C(↑)         LSE-D(↓)
>      Groundtruth                      5.45             8.52
>      Combined Encoder                 5.71             8.41
>      Hallo                            3.03             8.71
>      Audio2Head                       2.51            10.34
>      EAT                              4.39             9.35
>      SadTalker                        5.44            10.09
>      Proposed                         5.74             8.38
>
> Further, we experimented by interchanging the voice profile and the guided image to experiment with the bias, still, the performance was good. A few examples are added to the GitHub page.
>
> C6: At inference time, can the model overcome these potential biases? For example, if a child’s ..
>
> --Re: We have done experiments with having adult audio and child faces and vice versa and found that no bias is involved. We have added a few examples on the GitHub page.

---

> > ### Comment · Reviewer_4YGG · 2024-11-24
> >
> > Dear Authors,
> > Thank you for your response.
> > I have reviewed the reply you provided to my comments and those of the other reviewers.
> > The term "Audio Transformer Only" is unclear to me. From inspecting your code, it seems you are referring to the Prompt Guided Transformer, which would make the other one "Video Transformer Only," with both functioning as cross-attention modules. In this context, my question about the impact of vision tokens on audio generation remains. When you state, "from the above table, we can conclude," are you referring to "Audio Transformer Only"? If so, it appears that this module still utilizes audio-visual cross-attention, which inherently involves vision tokens.
> > If you argue that vision tokens play a vital role in audio generation, why is there no bias? For example, when a child’s voice is input alongside a man’s image, why doesn’t the generated audio reflect the characteristics of a man, given that the man’s facial features are cross-attended with the audio tokens in the Prompt Guided Transformer Encoder? If the cross-attention were restricted to lip features, this might align with your explanation, but since the entire face's features are used, this remains unclear. Thank you for sharing the qualitative examples, which suggest there is no bias. However, your responses seem contradictory. If vision tokens are critical to audio generation, how is bias avoided? This contradiction requires further clarification.
> > You agreed that my understanding of all previous works being re-evaluated is correct. While I acknowledge your justification for re-evaluating due to the need to trim the video to 10 seconds to fit your architecture, I still feel that including at least some experimental results directly from the original papers would have inspired greater confidence.
> > Finally, regarding the use of other pre-trained models, as most components are derived from existing pre-trained networks, my concern that this could have been framed as a zero-shot problem still stands. Your primary contribution seems to be the introduction of two cross-attention modules: the Prompt Guided Transformer Encoder (query = Audio, key/value = Vision, Text) and the Audio-Visual Transformer Encoder (query = Vision, key/value = Audio, Text). While the paper does not specify the number of layers in these modules, your code suggests they are implemented as single-layer architectures (cross-attention followed by self-attention). I am not fully convinced how this approach significantly differs from a zero-shot methodology.
> >
> > While I acknowledge this as an engineering effort to apply existing methods to address a real-world problem, I am not fully convinced that it demonstrates the level of novelty and experimental rigor expected for an ICLR paper.

---

> > > ### Author Response · Authors · 2024-11-25
> > > **Experimental results from previous work**
> > >
> > > Thanks for your further queries and suggestions for including results directly from the original papers. But in this context, we would like to point out that our experimental protocol is different. Moreover, we cannot experiment with our model with the individual protocol each previous work has adopted. Hence adopting the results from the previous paper is not relevant and will not be a fair comparison. Hence we have used a common protocol and datasets, which is used to evaluate the proposed and the previous work to bring in an unbiased comparison.

---

> > > ### Author Response · Authors · 2024-11-25
> > > **Bias in the model**
> > >
> > > Apology for any confusion. From your previous review “Note: If the output does not retain the childlike qualities of the reference audio, this indicates the model has learned a bias.” we consider that the model is biased if the contribution of both the modalities are not retained in the output; in other words, if both visual and audio tokens have their contribution in the output then there is no bias. Hence, we wanted to prove that the model is not biased to any of the modalities by showcasing the examples. For example with the voice of a child and the image of an adult, if the voice of an adult was generated we could have said that the model is biased by video modality.

---

> > > > ### Comment · Reviewer_4YGG · 2024-11-26
> > > >
> > > > **Response regarding use of pre-trained encoder and novelty:** My point is that the performance improvement primarily stems from the components you adapted from other pretrained works. Without the pre-trained weights, if you simply adopted the architectural components and trained them from scratch, the improvements wouldn't be significant, right? I don't see the addition of two cross-attention modules as a contribution substantial enough for an ICLR submission. Alternatively, you could have designed the experiments and written the paper to clearly demonstrate that adding these two cross-attention layers leads to a significant impact. However, that's not evident in the current presentation.
> > > >
> > > >
> > > > **Experimental results from previous work:** If each work introduces its own evaluation protocol and re-evaluates prior work using that protocol, do you think it’s a fair comparison? If this is the standard approach in the field, I’m okay with it.
> > > >
> > > >
> > > > **Bias in the model**: My concern is this, if the audio profile represents a child’s voice and the face image is of an adult man, the output should preserve the child’s voice and not alter the tone to match the man’s image. Now, consider another example: if the audio profile has a specific accent and the face image corresponds to a particular ethnicity, the output should retain the accent from the audio profile, not adapt it to the ethnicity implied by the image. If the output audio reflects the image’s ethnicity rather than the audio profile’s accent, that would be problematic.
> > > > While your examples suggest this issue isn’t occurring, it’s unclear why it wouldn’t happen, given that you apply cross-attention between face image tokens and audio tokens before audio generation. Either this cross-attention is having no significant effect, or there’s another factor at play that needs to be clarified.

---

> ### Author Response · Authors · 2024-11-25
> **Respose regarding use of pretrained encoder and novelty**
>
> Thanks again for your suggestions. To further clarify we like to mention that in our architecture, we have used the pre-trained encoder and decoders with the respective pre-trained weights which are further trained with the proposed architecture. Hence we cannot frame this as a zero-shot evaluation as the employed modules are not plug-and-play to accomplish the proposal. For example, instead of “HiFi Gan” we could have used any other existing Audio Encoder or defined an encoder that could have led to similar encoding.
>
>
> Further, we have mentioned in the manuscript (sections 3.1 and 4.1) that which parts from the existing models we have adopted, we are not “combining existing architectures trained for individual tasks”. For example, while we adopted the video diffusion pipeline part of “Hallo” we are not using the entire architecture. To illustrate further, we do not use the same AudioProcessor Hallo has used where audio was the driving modality, rather we use HifiGan and Wav2Vev in our pipeline as it is driven fully by the prompt text.
>
> Moreover, the beauty of using pre-trained weights is that it has existing learned representations which help the cross attentions to foster better exchange of information which is the important aspect in our proposal. In other words, the use of pre-trained encoders is beneficial while using the cross-attention in the transformers.
>
> As you have correctly pointed out, that we used a single-layer attention or shallow transformer, but still there was an exchange of information between different modalities as we used pre-trained weights. We have also experimented without using pre-trained weights but the results were not very promising.

---

> ### Author Response · Authors · 2024-11-27
> **Response regarding use of pre-trained encoder and novelty and**
>
> **Experimental results from previous work: **
>
> **Reply:** Thanks for your further insight and understanding that our experimental setup is different and cannot be directly mapped to SOTA.
>
>
> **---**My point is that the performance improvement primarily stems from the components you adapted from other pretrained works. Without the pre-trained weights, if you simply adopted the architectural components and trained them from scratch, the improvements wouldn't be significant, right?
>
> **Reply:** Thanks for your feedback. We would like to add that we have not adapted all the pre-trained weights of encoders and decoders directly from works such as Hallo or XXTS. Rather the generic respective pre-trained weights available were employed. For example, the HiFiGan encoder and generator were adopted from [3]. Moreover, the use of such pre-trained weights is quite popular for transfer learning and also in generation tasks, as established in works like ICLR 22[1], 23[2].
> Further, if we had learnt each encoder/decoder from scratch with large datasets for specific tasks independently, and then trained them with the whole architecture we would have achieved the same result but it would have been a time-consuming and impractical process (for example GPT 2 which is used as a decoder has 1.5 billion parameters and was trained on 8 million documents, from 45 million webpages).
>
>
> **__** I don't see the addition of two cross-attention modules as a contribution substantial enough for an ICLR submission. Alternatively, you could have designed the experiments and written the paper to clearly demonstrate that adding these two cross-attention layers leads to a significant impact. However, that's not evident in the current presentation.
>
> **Reply:**  The use of multiple latent features and entangling them to bring a meaningful learning representation for the personalized audio and video generation at the same time is the main contribution, and the cross attention only by itself has not given promising outputs. Furthermore, the whole learning strategy i.e. how to use the features from each encoder in a meaningful way and build the architecture for personalized AV generation is also novel and does not exist in the literature (as we have pointed out in the introduction section). Furthermore, for taking face generation such as in Hallo employed audio as the prompt, whereas the proposed model uses text as a prompt for both audio and video generation, which is a more complex task, that was achieved by the proposed architecture and the proposed latent representation.
>
> If any additional clarification is needed, we would be happy to discuss them further. Thank you again for your very helpful review of our work.
>
>
>
>
> 1.  	Geneface: Generalized And High-Fidelity Audio-Driven 3d Talking Face Synthesis
> 2.  	Latent Image Animator: Learning to Animate Images via Latent Space Navigation
> 3.  	HiFi-GAN: Generative Adversarial Networks for Efficient and High Fidelity Speech Synthesis

---

> ### Author Response · Authors · 2024-11-27
> **Bias in the model:**
>
> Thanks for further clarification, we understand your concern. Hopefully the following resolves the query.
> The primary goal of the model is to generate the audio as per the input audio profile and the video as per the source image along with the audio-visual synchronization. As we mentioned in the contribution at the end of the introduction, the primary novelty of the work is that “this is the first person-agnostic model which can foster a text-driven multimodal realistic audio-video synthesis that can be generalized to any identity “depending on the input image and audio profile.
> Why we do not find a bias: the model aims to learn the personal characteristics that are provided as inputs via the source image and reference audio profile
>
>
> As we mentioned (in the initial version of the submission) at the end of section 3.2, “the cross-attention mechanism enables the audio and video models to synchronize their outputs, ensuring that the generated audio and video components are temporally aligned”.
>
> To ensure that the proposed cross-attention does not add a bias, specific feature engineering by multi-latent entanglement is performed. As we can see in Fig 2, the encoded features from the prompt text and audio samples i.e. the output of the word2vec, HiFiGAN encoder and BPE are passed along with the cross attention from prompt guided transfer to the GPT2 decoder and then to HiFi generator to generate the voice profile specific audio output.  Similarly, for video pipeline output latent embedding on audio-visual features along with the visual tokens are passed to the reference net that can serve as a compact and compressed representation of facial animation sequences in the high-dimensional space which can be further decoded to get the source-image specific video. The choice of individual transformer blocks also helps to neutralise the bias (further illustrated with the help of the ablation).
>
> From the ablation, we can also find that biases is handled by the proposed model
>
> ·        Shared transformer encoders (STE) perform worse than the explicit transformer encoders (ETE) (ablation in Table 4 and Table 2 in supplementary)
>
> -As advised we have also done ablation with no video tokens in the Prompt Guided transformer the temporal comprehensiveness of the audio and synchronization matrix is worse
>
> -We can also observe that the sync matrix gets worse w/o attention.
>
> Thanks again for your valuable insight to enhance the quality of the paper, please let us know for any further concerns,

---

### Official Review · Reviewer_zqCR · 2024-11-04

**Soundness:** 3
**Presentation:** 2
**Contribution:** 2
**Rating:** 5
**Confidence:** 3

**Summary:**

This paper introduces a model designed for generating realistic, synchronized audio and video of a person speaking based on a static image, an audio profile, and a text prompt. The model uniquely integrates visual and audio features within a "multi-entangled latent space," allowing for synchronous facial animation and audio output that align with the speaker’s identity and the specified text. The model outperforms existing methods in video and audio quality, showing improvements in metrics such as FID for visual quality and MCD for audio quality.

**Strengths:**

1. The paper presents the proposed approach in a clear and comprehensible manner.
2. The method demonstrates strong performance across multiple benchmark datasets.
3. The accompanying code is provided, facilitating reproducibility and further analysis.

**Weaknesses:**

1. The paper claims that all model checkpoints and the proposed dataset are provided via a GitHub link. However, I was unable to locate these resources.
2. The clarity of several figures, such as Fig. 1, Fig. 3, and Fig. 4, could be improved. The current quality detracts from the overall readability and reader experience.
3. The paper lacks essential technical details, such as the selection of hyperparameters, the number of training iterations, training time, and the balance of loss functions.

**Questions:**

Please refer to Weaknesses.

---

> ### Author Response · Authors · 2024-11-21
> **Author response**
>
> We sincerely thank the reviewer for providing helpful feedback on our work. Based on that feedback, we will added the following to our revised paper:
>
> C1: The paper claims that all model checkpoints and the proposed dataset are provided via a GitHub link. However, I was unable to locate these resources.
>
> --Re: Sorry for the confusion. We have added the checkpoint and some data on the GitHub page. The full dataset used for experiments will be published after the acceptance of the paper.
>
>
> C2: The clarity of several figures, such as Fig. 1, Fig. 3, and Fig. 4, could be improved. The current quality detracts from the overall readability and reader experience.
>
> --Re: Thanks for the suggestion we will improve the figure and add it to the revised version of the paper.
>
> C3: The paper lacks essential technical details, such as the selection of hyperparameters, the number of training iterations, training time, and the balance of loss functions.
>
> ---Re: We have trained the models for 10 epochs, with a batch size of 8, and 1900 iterations.  The loss functions were balanced as follows: joint_loss = hy*audio_loss + video_loss, where hy=0.1

---

### Official Review · Reviewer_9b4k · 2024-11-04

**Soundness:** 2
**Presentation:** 2
**Contribution:** 2
**Rating:** 5
**Confidence:** 2

**Summary:**

The paper presents a novel approach for generating realistic speaking faces by synthesizing a person's voice and facial movements from a static image, a voice profile, and a target text. The model uses a multi-entangled latent space to integrate text, image, and audio data for synchronous audio-video generation. The proposed architecture consists of three phases: encoding, multi-entangled latent space, and decoding. The model demonstrates superior performance compared to state-of-the-art methods across various datasets.

**Strengths:**

(1) The use of a multi-entangled latent space for integrating multiple modalities is novel and addresses existing limitations in synchronous audio-video generation. (2) The three-phase model (encoding, latent space, decoding) ensures effective processing and generation of audio-visual content. (3) The model shows superior performance on multiple datasets, indicating its robustness and generalization capability.

**Weaknesses:**

(1) The multi-entangled latent space and cross-modal attention mechanisms seem quite complex. More ablation studies are needed to analyze their individual contributions. (2) Computational cost and generation time are not discussed. The multi-stage framework with transformers and diffusion models may be expensive to train and slow during inference. (3) Dependence on a clean audio profile of the target identity may limit applicability in certain scenarios. Robustness to noise needs to be evaluated.

**Questions:**

(1) How does the model handle diverse accents and languages in the audio input?
(2) What are the computational requirements for training and deploying this model?
(3) Can the model be adapted for real-time applications, and if so, what are the latency implications?

---

> ### Author Response · Authors · 2024-11-21
> **Author response**
>
> We sincerely thank the reviewer for providing helpful feedback on our work. Based on that feedback, we will added the following to our revised paper:
>
> (C1) The multi-entangled latent space and cross-modal attention mechanisms seem quite complex. More ablation studies are needed to analyze their individual contributions.
>
> --Re: Thanks for your insight. We have done additional ablation suppressing the encoders and latent entanglements. To see the individual contributions of the Audio and Video attention mechanisms we ran the experiment removing the attention for the other modality (Video Transformer Encoder only, Audio Transformer Encoder only). Further, for encoder-based ablation we ran an experiment removing HiFi-Gan i.e. the speaker embedding as a feature - this led to the final result having a generic voice profile. We also ran an experiment removing the Wav2Vev encoder. The following are the results. We can note that both of the features are important for generation.
>
>                             FID (↓)           FVD (↓)          FVMD (↓)           FAD Score (↓)       MCD (↓)          STOI (↑)
>     Video Transformer only     68.31	   78.42	   5747.04	        304.98	           81.17	    0.13
>     Audio Transformer only     69.02	   79.35	   6576.85	        301.49	           80.65	    0.13
>     No Hifigan                 85.25	   94.28	    7483.4	        498.33 	           87.51	    0.09
>     No wav2vec                 70.1	           80.96	   5926.64 	         309.95	           89.58	    0.11
>     Proposed                  42.88            49.78            4192.07              241.75            75.39            0.17
>
>
>  (C2) Computational cost and generation time are not discussed. The multi-stage framework with transformers and diffusion models may be expensive to train and slow during inference.
>
> --Re: The parameter details for the transformer used is discussed in the supplementary. The total parameter size of the model is 1,575,936 M. The model does 5.39 GFLOPs for the whole generation pipeline. The generation time for a 10-second is around 2 seconds. The inference needs 12GB of VRAM.
>
>
> (C3) Dependence on a clean audio profile of the target identity may limit applicability in certain scenarios. Robustness to noise needs to be evaluated.
>
> --Re: We have done experiments by blurring the audio but still the model is found to be effective. We have added a few examples in the GitHub page.
>
>
> Questions:
> (C1) How does the model handle diverse accents and languages in the audio input?
>
> --Re: We have done experiments with varying accents with kids and the results found to be effective. We have added a few examples in the github page.
>
>
> (C2) What are the computational requirements for training and deploying this model?
>
> --Re:  The model inference needs 12GB of VRAM.
>
>
> (C3) Can the model be adapted for real-time applications, and if so, what are the latency implications?
>
> ---Re: The generation time for a 10-second is around 2 seconds. Hence we believe it can be applied for real-time applications.

---

### Official Review · Reviewer_wRsR · 2024-11-06

**Soundness:** 2
**Presentation:** 3
**Contribution:** 3
**Rating:** 6
**Confidence:** 3

**Summary:**

This paper presents a novel framework for jointly generating the visual and audio cues of talking faces based on a text prompt, an image, and an audio profile. This work addresses issues such as poor lip synchronization and limited expressiveness in previous approaches which synthesize the visual and audio cues separately. This approach introduces a multi-entangled latent space to align audio-visual elements. The model utilizes transformers and diffusion models to achieve precise, person-agnostic synchronization of voice and facial expressions. Quantitative results of this work demonstrates favorable performance over baselines across multiple datasets.

**Strengths:**

1. Synthesizing audio and video together based on text input addresses a meaningful setting in talking faces generation.
2. The framework design appears reasonable, employing attention mechanisms for effective cross-modal fusion, and the presentation is overall clear.
3. The quantitative results suggest a notable improvement in quality over recent approaches, with favorable scores in metrics like FVD, FID, and FAD.
4. The ablation study shows that key techniques are effective.

**Weaknesses:**

1. The experiments seems to lack a dedicated evaluation for audio-visual synchronization, which is identified as a problem this paper aims to address.
2. Without qualitative results, such as playable video and audio samples, it’s challenging to fully assess and compare critical aspects like coherence, smoothness, synchronization, realisticness and expressiveness. The two examples provided in the GitHub link are a helpful start, but a broader range of samples would provide a more comprehensive evaluation.

**Questions:**

1. The framework consists of many modules. Could you elaborate which of these components use pretrained weights, and which are being trained?
2. Fig.3 and Fig.4 presents frames from generated videos. Could you explain how were these frames sampled?

**Details Of Ethics Concerns:**

As with all advancements in talking-face generation, this technology carries the potential for misuse in creating synthetic / misleading content, depending on the user’s intent.

---

> ### Author Response · Authors · 2024-11-21
> **Author response**
>
> We sincerely thank the reviewer for providing helpful feedback on our work. Based on that feedback, we will added the following to our revised paper:
>
> C1: The experiments seems to lack a dedicated evaluation for audio-visual synchronization, which is identified as a problem this paper aims to address. Without qualitative results, such as playable video and audio samples, it’s challenging to fully assess and compare critical aspects like coherence, smoothness, synchronization, realisticness and expressiveness. The two examples provided in the GitHub link are a helpful start, but a broader range of samples would provide a more comprehensive evaluation.
>
> Pls add more to github with different variation for ex indian individual, old subjects etc
>
> -------Re: Thank you for your valuable insight. From the literature, it can be found that lip synchronization is a good means for evaluation of audio-visual synchronization (along with critical aspects like coherence, smoothness, synchronization, realisticness and expressiveness). We did experimentation with two metrics as mentioned in Wav2Lip [1]. The first is the average error measure calculated in terms of the distance between the lip and audio representations, “LSE-D" (“Lip Sync Error - Distance"). A lower LSE-D denotes a higher audio-visual match, i.e., the speech and lip movements are in sync. The second metric is the average confidence score, “LSE-C" (Lip Sync Error - Confidence). The higher the confidence, the better the audio-video correlation. A lower confidence score denotes that there are several portions of the video with completely out-of-sync lip movements. From the below table, we can conclude that our proposed model has performed better than the other SOTA models and is close to the ground truth because we generate both modalities.
>
> 1: Prajwal, K R and Mukhopadhyay, Rudrabha and Namboodiri, Vinay P. and Jawahar, C.V. A Lip Sync Expert Is All You Need for Speech to Lip Generation In the Wild. Proceedings of the 28th ACM International Conference on Multimedia
>
>                                       LSE-C(↑)         LSE-D(↓)
>      Groundtruth                      5.45             8.52
>      Combined Encoder                 5.71             8.41
>      Hallo                            3.03             8.71
>      Audio2Head                       2.51            10.34
>      EAT                              4.39             9.35
>      SadTalker                        5.44            10.09
>      Proposed                         5.74             8.38
>
> Further, as per request, we have added more examples in GitHub page with older and younger subjects and different geographical demographics.
>
> C2: Questions: The framework consists of many modules. Could you elaborate which of these components use pretrained weights, and which are being trained?
>
> ----Re: Thanks for your interest. The Hifi-Gan, Wav2Vec Encoders, the Variational Autoencoder, Diffusion Models, and the GPT2 Decoder are pre-trained. Further, we’ve fine-tuned them with our multi-latent entanglement with the transformer to achieve the proposed learning representation of simultaneous and seamless audio-video generation based on the prompt text. It is important to note that both the transformer encoders are trained from scratch along with the above-mentioned pre-trained layers.
>
> C3: Fig.3 and Fig.4 presents frames from generated videos. Could you explain how were these frames sampled?
>
> -----Re: These frames were sampled at equal intervals across the videos whose frames were used. Frame 1, Frame 25, Frame 50, Frame 75 and Frame 100 were sampled.

---

### Author Response · Authors · 2024-11-23
**Generaic response**

Hi Everybody,

We kindly thank the reviewers for their helpful feedback on our work. We appreciate that they find our approach effective (wRsR, 9b4k, zqCR and RNFb). We have submitted our revised version addressing all the concerns raised (all changes are highlighted in red colour text). Kindly take some time to have a look at the revised version, additional information in the GitHub page and the response as open review comments. Feel free to reply for any further clarification.

Best

Playing For You Authors

---

### Author Response · Authors · 2024-11-26

Dear Reviewers,

Thanks again for your great efforts and valuable review. Your insight has helped to improve the soundness of our work. We have carefully addressed all concerns and provided detailed responses to each reviewer. We hope you will find the responses satisfactory. As the end of the rebuttal phase approaches, we would be grateful if you could share your feedback regarding our response. We will be pleased to clarify any further concerns.

Thanks in advance,

Paper 10887 authors

---

> ### Author Response · Authors · 2024-11-28
> **Changing highlightment in rubuttal file**
>
> Dear All,
>
> As the end of the rebuttal phase is approaching, we have changed the colour of the revised parts (which were previously marked in red) in the manuscript to black.
>
> For your convenience, you can still easily access the previously revised version at this link (https://github.com/Playing-for-you/Playing-for-you/blob/main/Revised_Paper.pdf) and (https://github.com/Playing-for-you/Playing-for-you/blob/main/Supplementary.pdf), in which the revised parts are highlighted.
>
> Paper 10887 authors

---

### Author Response · Authors · 2024-12-04
**Author response summary**

As we approaching the verge of discussion period, we want to sincerely thank the reviewers again for their valuable feedback on our work.

Below, we summarize the major changes made to our paper based on the reviewer feedback. All of the changes to the paper are marked in red in the GitHub page of the project.https://github.com/Playing-for-you/Playing-for-you

1. Rigorous ablation:  detailed ablation while replacing the latents and encoding combination and different variations of entanglement.

2. Audio video synchronisation matrix, along with experiment on it.

3. More precise details of experiment, ethical statement, also we explained the angle of novelty.


We believe these changes have substantially address the concerns from the reviewers and improved our paper, which we are very grateful for.

Thank you all again for your.

Regards
Authors of Playing for You

---

### Meta-Review · Area_Chair_xjfJ · 2024-12-15

**Metareview:**

The approach demonstrates potential for real-world applications and shows measurable performance improvements in experiments. However, the technical contributions are limited, relying on existing architectures, and the paper lacks sufficient ablation studies to justify the proposed components. Additionally, there is no discussion on resource efficiency or scalability, and the "multi-entangled latent space" concept is not well-explained. Lip synchronization is mentioned but not thoroughly evaluated, and ethical concerns regarding misuse in generating deepfakes are not addressed.

The majority of reviewers recommended rejection. After reviewing the paper and rebuttal, the Area Chair also concluded that the paper should be rejected.

**Additional Comments On Reviewer Discussion:**

The reviewer appreciates the engineering effort but questions the novelty, as the improvements seem to stem from adapting pre-trained components, and the addition of cross-attention modules does not appear substantial enough for an ICLR submission. They also raise concerns about the fairness of re-evaluating previous work with a new evaluation protocol. Additionally, the reviewer asks for clarification on how the cross-attention mechanism ensures the preservation of audio characteristics (e.g., accent or voice tone) when applied to images, as this could introduce bias.

---

### Decision · Program_Chairs · 2025-01-22

Reject